# Diagnostic and Prognostic Roles of GATA3 Immunohistochemistry in Urothelial Carcinoma

**DOI:** 10.3390/medicina59081452

**Published:** 2023-08-11

**Authors:** Daeseon Yoo, Kyueng-Whan Min, Jung-Soo Pyo, Nae Yu Kim

**Affiliations:** 1Department of Urology, Daejeon Eulji University Hospital, Eulji University School of Medicine, Daejeon 35233, Republic of Korea; dsyoo@eulji.ac.kr; 2Department of Pathology, Uijeongbu Eulji Medical Center, Eulji University School of Medicine, Uijeongbu-si 11759, Republic of Korea; kyueng@eulji.ac.kr; 3Department of Internal Medicine, Uijeongbu Eulji Medical Center, Eulji University School of Medicine, Uijeongbu-si 11759, Republic of Korea

**Keywords:** urothelial carcinoma, GATA3, immunohistochemistry, overall survival, meta-analysis

## Abstract

This study aimed to evaluate the diagnostic and prognostic roles of GATA-binding protein 3 (GATA3) immunohistochemistry in urothelial carcinoma (UC) using a meta-analysis. We investigated GATA3 immunohistochemical expression rates and performed a subgroup analysis based on tumor site, study location, and histological subtypes. The overall survival rates of patients with GATA3-positive and -negative UC were compared. The estimated GATA3 expression rate was 0.748 (95% confidence interval [CI]: 0.704–0.787). GATA3 expression rates in the urinary bladder and urinary tract were 0.775 (95% CI: 0.727–0.818) and 0.614 (95% CI: 0.426–0.774), respectively. The GATA3 expression rates of noninvasive and invasive UCs were 0.965 (95% CI: 0.938–0.980) and 0.644 (95% CI: 0.581–0.702), respectively. In invasive UCs, there was a significant difference in GATA3 expression between non-muscular invasion and muscular invasion subgroups (0.937, 95% CI: 0.883–0.967 vs. 0.753, 95% CI: 0.645–0.836). GATA3 expression was the highest in the microcytic subtype among the histologic subtypes (0.952, 95% CI: 0.724–0.993). There was a significant correlation between GATA3 expression and better prognosis (hazard ratio: 0.402, 95% CI: 0.311–0.521). Taken together, GATA3 expression significantly correlated with low-stage and better prognosis in UC. GATA3 expression is highly variable across histological subtypes, and one should be careful while interpreting GATA3 expression.

## 1. Introduction

Urothelial carcinoma (UC) is a common malignancy of the urinary tract that affects approximately 430,000 people and causes 165,000 deaths worldwide annually [1]. UC is classified into noninvasive and invasive and can be sub-staged according to the presence of muscle invasion. The prognosis differs depending on the extent of UC, which in turn influences treatment decisions. Muscle-invasive UC is an aggressive and potentially life-threatening form of cancer. Thus, the identification of reliable biomarkers for invasive UC can help in the early detection and risk stratification of patients, leading to personalized treatment strategies and improved clinical outcomes [2,3]. GATA-binding protein 3 (GATA3) has emerged as a promising biomarker for UC. GATA3 is a transcription factor that regulates cell differentiation and proliferation and is commonly expressed in urothelial and breast epithelial cells [4,5]. GATA3 plays a role in regulating luminal differentiation in the breast epithelium [6,7]. In addition, GATA3 may be involved in the development or maintenance of various tissues, such as the skin, hair shafts [8,9], and endothelial cells of great vessels [10]. In breast cancer, GATA3 is associated with tumor differentiation and recurrence [11,12]. Many studies have suggested that GATA3 plays an important role as a tumor suppressor in the prevention of urothelial cancer progression and metastasis [13,14,15,16]. However, detailed GATA3 expression in the histological subtypes of UC remains poorly understood. Additionally, the prognostic value of GATA3 expression remains unclear [13,14,15,16,17,18,19,20,21,22,23,24,25,26,27,28,29,30,31,32,33,34,35,36,37,38,39,40,41,42,43,44,45,46,47,48,49,50,51,52,53,54]. It may be difficult to obtain biopsied tissues of sufficient depth, which may limit the assessment of invasiveness. If GATA3 immunohistochemical (IHC) expression differs between tumor grades, it could be useful in assessing biopsy specimens. This study aimed to evaluate the diagnostic and prognostic roles of GATA3 IHC in UC using meta-analysis. GATA3 IHC expression rates were investigated and a meta-analysis was performed. In addition, a subgroup analysis was performed based on the tumor site, study location, tumor stage, and histologic subtypes. The overall survival rates of patients with GATA3-positive and -negative UC were compared.

## 2. Materials and Methods

### 2.1. Literature Search and Selection Criteria

Relevant articles were obtained by searching the PubMed and MEDLINE databases up to 15 April 2023. The search was performed using ‘urothelial carcinoma’, ‘GATA3’, and ‘immunohistochemistry’ as search terms. The titles and abstracts of all the returned articles were screened for exclusion. Review articles were screened to identify the eligible studies. Studies in English language addressing GATA3 expression in human UC were included. Case reports and review articles were excluded.

### 2.2. Data Extraction

Thirty-nine articles were included and reviewed in this meta-analysis [15,17,18,19,20,21,22,23,24,25,26,27,28,29,30,31,32,33,34,35,36,37,38,39,40,41,42,43,44,45,46,47,48,49,50,51,52,53,54]. From eligible studies, we collected the following information: first author’s name, publication date, study location, number of patients analyzed, tumor site, and expression rates of GATA3. For the quantitative aggregation of survival results, the correlation between GATA3 expression and the survival rate was analyzed according to the hazard ratio (HR). Because the survival data were in the form of graphical representations of survival distributions, survival rates were extracted at specified times to reconstruct the HR estimate and its variance under the assumption that patients were censored at a constant rate during the time intervals [55]. The published survival curves were independently evaluated by two authors to reduce variability. The HRs were then combined into an overall HR using Peto’s method [56]. Disagreements were resolved by consensus.

### 2.3. Statistical Analyses

To perform the meta-analysis, all data were analyzed using the Comprehensive Meta-Analysis software package (Biostat, Englewood, NJ, USA). The immunohistochemical expression of GATA3 in UC was investigated in eligible studies. In addition, subgroup analyses based on tumor location, study location, tumor stage, and histological subtype were performed. Correlation between GATA3 expression and overall survival was also evaluated. Because the eligible studies used different antibodies and evaluation criteria for different populations, the random-effects model was more suitable than the fixed-effects model to interpret the results. Heterogeneity and sensitivity analyses were conducted to assess the heterogeneity of eligible studies and the impact of each study on the combined effect. Heterogeneity between studies was checked using Q and I^2^ statistics, and *p*-values were calculated. In addition, the significance of difference between subgroups was evaluated using a meta-regression test. Begg’s funnel plot and Egger’s test were used to assess publication bias. Statistical significance was set when *p* < 0.05.

## 3. Results

### 3.1. Selection and Characteristics of Studies

Database search was performed using the above-mentioned keywords. One hundred fifty-nine reports were identified in the database search. Articles were selected by screening the titles and abstracts. During the screening and full-text review, 38 articles were excluded because of insufficient information. In addition, 57 articles were excluded because they focused on other diseases. The remaining 25 articles were excluded because they were non-original (*n* = 22), used animals or cell lines (*n* = 2), or were not written in English (*n* = 1). Finally, 39 articles were included in the meta-analysis (Figure 1 and Table 1).

### 3.2. Meta-Analysis of GATA3 IHC Expression in Urothelial Carcinoma

The GATA3 IHC expression rate was 0.748 (95% confidence interval [CI]: 0.704−0.787) in all cases (Table 2). The expression rates of GATA3 in the urinary bladder and urinary tract were 0.775 (95% CI: 0.727−0.818) and 0.614 (95% CI: 0.426−0.774), respectively. The GATA3 expression rate in UC was significantly higher in the urinary bladder than in the urinary tract (*p* = 0.001 in the meta-regression test). In the subgroup analysis based on study location, GATA3 expression rates were 0.741 (95% CI: 0.627−0.829), 0.748 (95% CI: 0.590−0.859), 0.775 (95% CI: 0.723−0.819), and 0.909 (95% CI: 0.561−0.987) in the American, Asian, European, and Oceania subgroups, respectively. The GATA3 expression rates in noninvasive and invasive UC were 0.965 (95% CI: 0.938−0.980) and 0.644 (95% CI: 0.581−0.702), respectively (Table 3). The GATA3 expression rate in carcinoma in situ was 0.956 (95% CI 0.878−0.985). GATA3 expression was significantly lower in muscle-invasive UC than in non-muscle-invasive UC (0.753, 95% CI: 0.645−0.836 vs. 0.937, 95% CI: 0.883−0.967; *p* = 0.041 in the meta-regression test). In a subgroup analysis based on histological subtypes, clear cell, microcystic, and pleomorphic adenocarcinoma subtypes showed GATA3 expression of more than 90%. The lymph node metastasis rate was not significantly different between GATA3-positive and -negative UCs (0.375, 95% CI: 0.282−0.478 vs. 0.340, 95% CI: 0.239−0.459). We evaluated the prognostic implications of GATA3 expression in UC. UC patients with GATA3 expression had better overall survival than those without GATA3 expression (hazard ratio 0.402, 95% CI: 0.311−0.521; Figure 2). However, there was no significant correlation between GATA3 expression and overall survival in UCs with genetically unstable luminal type.

## 4. Discussion

Previously, the prognostic implications of various parameters have been investigated. Basically, UC with higher tumor stage was significantly correlated with worse survival [57]. Tumor depth in UC is classified according to subepithelial and muscular invasion. This stratification may be important in predicting prognosis and selecting treatment for UC. Routine serum markers were studied as predictive markers of prognosis in UC patients [57]. There were also significant correlations between worse prognosis and high C-reactive protein and low hemoglobin [57]. In addition, patient’s age could affect survival of UC patients with radical cystectomy [58]. GATA3 is expressed in various tissues and cancers [59]. GATA3 is involved in the differentiation of various cells, including the urothelium, breast epithelium, and T lymphocytes [59]. GATA3 is involved in the luminal differentiation of breast epithelium through gene regulation of MUC1/EMA [59]. In addition, GATA3 is correlated with the development or maintenance of skin and skin appendages and endothelial cells [59]. In non-muscle-invasive UC, intravesical immunotherapy with Bacillus Calmette–Guérin (BCG) is used to reduce recurrence [60]. However, despite the use of BCG, tumor recurrence is common. Group 2 innate lymphoid cells (ILC2) express high levels of the transcription factor GATA3 [60]. As ILC2 may be the immunotherapeutic target in UC, elucidation of the role of GATA3 may be important.

Although several studies have reported the diagnostic and prognostic roles of GATA3 in malignant tumors, the results vary significantly according to tumor type. Additionally, there have been several studies on GATA3 expression in UC; however, individual studies may be insufficient to assess the significance of GATA3 expression in UC. Therefore, it is necessary to obtain and analyze integrated information through systematic reviews, including meta-analyses. To the best of our knowledge, this is the first meta-analysis to elucidate the clinicopathological implications of GATA3 IHC markers in UC.

GATA3 promotes cell proliferation and differentiation in many tissues and cell types [61,62,63]. GATA3 is also involved in malignant progression [63,64]. Some cancers showed GATA3 overexpression, such as breast and colorectal carcinoma [63,64]. In breast cancer, GATA3 induces tumor differentiation in undifferentiated carcinomas [65]. Carcinomas with high GATA3 expression include basal cell carcinoma, breast cancer, germ cell tumors, and low-grade UC (>90 percent) [59]. Conversely, carcinomas with low GATA3 expression are positive in less than 10% of cases, including gastric and colorectal adenocarcinoma, endometrial adenocarcinoma, hepatocellular carcinoma, cholangiocarcinoma, prostatic adenocarcinoma, and thyroid carcinoma [59]. GATA3 is expressed in primary UC but also in metastatic UC. Naik et al. reported GATA3 expression in all metastatic UC [41]. Efforts have been made to classify UC based on GATA3 expression levels. UC can be divided into luminal and basal subtypes, with GATA3 expression in the luminal subtype and cytokeratin 5/6 (CK5/6) expression in the basal subtype. Budina reported that the luminal subtype with GATA3 expression is the predominant phenotype in T1 cancers [25]. In contrast, the basal subtype with no GATA3 expression accounts for only 5% of all T1 cancers [25]. However, there were no significant differences in sex, ethnicity, grade, recurrence, or survival between the luminal and non-luminal phenotypes [25]. Seiler et al. reported that the classification of luminal and basal tumors affects the efficacy of cisplatin-based neoadjuvant chemotherapy in muscle-invasive UC [21]. Therefore, GATA3 IHC is useful for differentiating UC phenotypes and for the differential diagnosis between UC and other malignant tumors.

We investigated the expression of GATA3 under various conditions and performed a meta-analysis. GATA3 expression was significantly lower in the urinary tract than in the urinary bladder (0.614; 95% CI: 0.426−0.774 vs. 0.775; 95% CI: 0.727−0.818). GATA3 expression in UC can differ significantly depending on the invasion status. Interestingly, even in invasive UC, GATA3 expression significantly differed depending on the presence or absence of muscle invasion. There was no significant difference between noninvasive and non-muscle-invasive UC (0.965, 95% CI: 0.938−0.980 vs. 0.937, 95% CI: 0.883−0.967). Based on this result, the presence of muscle invasion is an important factor in differentiating GATA3 expression, rather than simply the presence of invasion. Although GATA3 expression is grouped with Ta and T1 and T2 and above, the GATA3 expression rate is also significantly higher in muscle-invasive UC, with a rate of 0.753 (95% CI: 0.645−0.836). Taken together, GATA3 can be a useful marker for differentiation from muscle-invasive UC in small and fragmented tissues. Because the stratification may be important in predicting prognosis and selecting treatment for UC. Mohammed et al. reported that high GATA3 expression is associated with significantly larger tumor size in UC [14]. In our study, the lymph node metastasis rate was not significantly different between GATA3-positive and -negative UCs. Our results suggest that the GATA3 expression is associated with a better prognosis, which may be related to lower staging. In contrast to our result, there was no significant difference in survival between GATA3-positive and -negative UC [14]. However, they only included the invasive UC [14]. In our study, only invasive cases were included in the meta-analysis of survival [33,43]. Meta-analyses are more meaningful because they can show different results depending on the study. Unlike Mohammed’s report, our result showed the comparison of GATA3 expression in both noninvasive and invasive UCs.

Miettinen et al. reported GATA3 expression in various epithelial and non-epithelial tumors [59]. In breast cancer, the ductal and lobular carcinomas showed higher GATA3 expression rates (92% and 100%, respectively) [59]. However, some tumors showed very low GATA3 expression rates. Neuroendocrine tumors, including small cell lung cancer and Merkel cell carcinoma, showed no cases with GATA3 expression [59]. Some carcinomas, such as colorectal adenocarcinoma, hepatocellular carcinoma, cholangiocarcinoma, gastric adenocarcinoma, lung adenocarcinoma, and papillary and follicular thyroid carcinoma, have less than 10% of GATA3 expression rates [59]. Due to its location, UC may need to be differentiated from kidney or prostate cancer in some cases. There may also be high-grade UC that needs to be differentiated from metastatic squamous cell carcinoma of the uterine cervix and anal canal. There may be immunohistochemical markers that are specific for some cancers, but in some cases they may be less useful. As GATA3 expression may vary among histological subtypes of UC, its use in differentiating from other tumors may have limitations. A previous report showed that among renal cell carcinomas, chromophobe had 51% GATA3 expression, and the remaining subtypes had 2% expression [59]. In addition, prostatic adenocarcinoma was positive for GATA3 in 2% of the cases [59]. In the kidney, except for chromophobe renal cell carcinoma, there is a low rate of GATA3 positivity, which may be helpful for differentiation. Like our result, Mohammed et al. reported the diagnostic role of GATA3 in differentiation between UC and renal cell carcinoma and prostatic adenocarcinoma [14].

We investigated GATA3 expression in the different histological subtypes of UC. Among various histological subtypes, conventional UC is the most common. In our meta-analysis, the subtypes of UC with GATA3 expression in the overall invasive UCs were adenocarcinoma, adenoid differentiation, lymphoepithelioma-like, sarcomatoid, small cell neuroendocrine carcinoma, squamous cell carcinoma, squamous differentiation, signet ring cell carcinoma, and undifferentiated subtypes. Of these, adenocarcinoma, small-cell neuroendocrine carcinoma, and squamous cell carcinoma have GATA3 expression levels of less than 20%. UC with glandular differentiation, lymphoepithelioma-like UC, sarcomatoid UC, and signet ring cell carcinoma had GATA3 expression rates of 47.4%, 30.0%, 40.7%, and 40.9%, respectively. Clear cell, microcystic, and pleomorphic adenocarcinoma subtypes showed GATA3 expression of more than 90%. Interestingly, undifferentiated UC showed GATA3 expression rate of 64.3%. Different histological subtypes exhibited varying GATA3 expression levels. Subtypes with significant differences in GATA3 expression may be useful when differentiation is difficult in small tissues. In the present study, GATA3 expression was evaluated in urothelial carcinoma in situ. The GATA3 expression rate was 0.956 (95% CI 0.878−0.985). There was no significant difference between urothelial carcinoma in situ and UC with subepithelial invasion. Urothelial carcinoma in situ is a non-muscle-invasive UC and may progress to an invasive UC. Therefore, GATA3 may be useful for differentiation from an invasive lesion.

This study has several limitations. First, a total of 55 subsets were included in the invasive UC subgroup, and 19 subsets were included in the non-muscle-invasive and muscle-invasive UC subgroups. As the remaining 36 subsets were not classified according to muscle invasion, the analysis was limited. Second, there may be prognostic differences based on histological subtypes. However, a meta-analysis of the prognostic role of GATA3 within histological subgroups could not be performed because of insufficient information. Third, the detailed analysis based on patients’ age could not be performed due to insufficient information.

## 5. Conclusions

GATA3 expression is significantly higher in noninvasive and non-muscle-invasive UC than in muscle-invasive UC. In addition, GATA3 expression significantly correlated with improved overall survival in UC. GATA3 expression is highly variable across histologic subtypes. Furthermore, GATA3 IHC is useful in differentiating between UC subtypes and between low- and high-grade UCs.

## Figures and Tables

**Figure 1 medicina-59-01452-f001:**
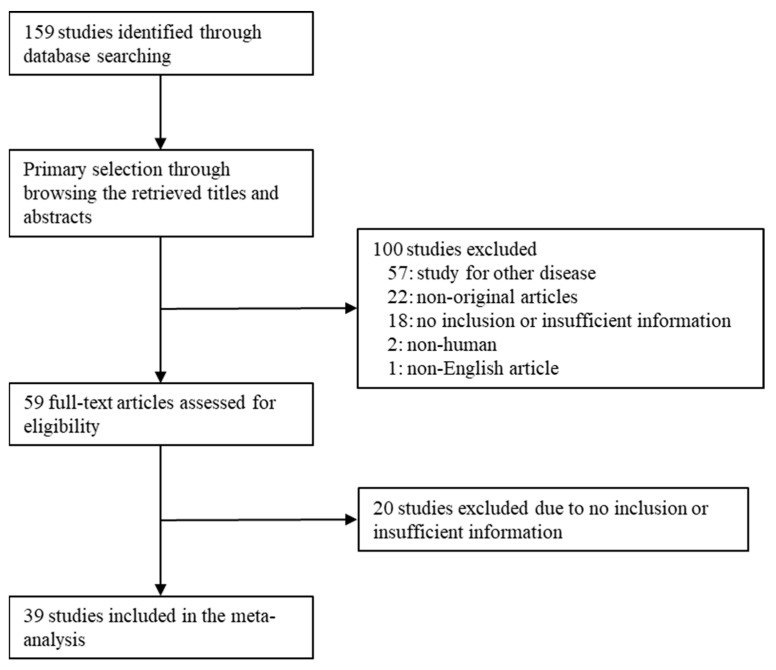
Flow chart of the search strategy.

**Figure 2 medicina-59-01452-f002:**
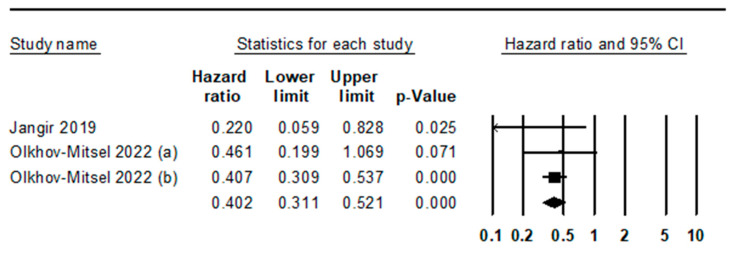
Forest plot for the correlation between GATA3 expression and overall survival. (a) Genetically unstable cases of luminal UC and (b) urothelial-like cases of luminal UC, [33,43].

**Table 1 medicina-59-01452-t001:** Main characteristics of the eligible studies.

First Author	Location	Organ	Number of Patients	First Author	Location	Organ	Number of Patients
Agaimy 2016 [17]	NA	UB, UT	14	Liang 2014 [37]	USA	UB	127
Agarwal 2019 [18]	India	UB	74	Lin 2014 [38]	USA	UB	98
Barth 2018 [19]	Germany	UB	156	Liu 2012 [39]	USA	UB	72
Bertz 2020 [20]	Germany	UB	18	Lopez Beltran 2014 [40]	NA	UB	20
Bontoux 2021 [21]	France	UB	184	Naik 2021 [41]	India	UB, UT	122
Borhan 2017 [22]	USA	UB	45	Oh 2016 [42]	Korea	UB	138
Bruch 2023 [23]	Germany	UB	2406	Olkhov-Mitsel 2022 [43]	Canada	UB	243
Brunelli 2022 [24]	Italy	UB	117	Paner 2014 [44]	Various	UB	111
Budina 2022 [25]	USA	UB	67	Perrino 2019 [45]	USA	UT	26
Chang 2013 [26]	USA	UT	56	Plage 2022 [15]	Germany/Poland	UB	2636
Ellis 2013 [27]	USA	UB	49	Reiswich 2023 [46]	Germany	NA	1066
Gonzalez-Roibon 2013 [28]	USA	UT	25	Samaratunga 2016 [47]	New Zealand	UB	11
Gulmann 2013 [29]	USA	UT	85	Sanfrancesco 2016 [48]	USA	UB	26
Guo 2020 [30]	USA	UB	72	Tian 2015 [49]	USA	UB	278
Haghayeghi 2021 [31]	USA	UB	42	Verduin 2016 [50]	USA	UB	86
Inoue 2017 [32]	USA	UT	48	Wang 2017 [51]	USA	UT	17
Jangir 2019 [33]	India	UB	40	Weyerer 2019 [52]	Germany	UB	55
Kim 2020 [34]	Korea	UB	166	Zhao 2013 [53]	USA	UB	69
Leite 2022 [35]	Brazil	UB	25	Zinnall 2018 [54]	Germany	UB	91
Leivo 2016 [36]	USA	UB	89				

UB, urinary bladder; UT, urinary tract.

**Table 2 medicina-59-01452-t002:** Estimated GATA3 positive rates in urothelial carcinomas.

	Number of Subsets	Fixed Effect [95% CI]	Heterogeneity Test (*p*-Value)	Random Effect [95% CI]	Egger’s Test (*p*-Value)
Overall	38	0.726 [0.716, 0.735]	<0.001	0.748 [0.704, 0.787]	0.694
Tumor site					
Urinary bladder *	29	0.731 [0.720, 0.741]	<0.001	0.775 [0.727, 0.818]	0.395
Urinary tract	6	0.538 [0.477, 0.599]	<0.001	0.614 [0.426, 0.774]	0.391
Study location					
America	20	0.670 [0.641, 0.697]	<0.001	0.741 [0.627, 0.829]	0.215
Asia	4	0.707 [0.663, 0.747]	<0.001	0.748 [0.590, 0.859]	0.130
Europe	9	0.738 [0.727, 0.749]	<0.001	0.775 [0.723, 0.819]	0.416
Oceania	1	0.909 [0.561, 0.987]	<0.001	0.909 [0.561, 0.987]	-

CI, Confidence interval. *, *p* = 0.011 in the meta-regression test between urinary bladder and urinary tract subgroups.

**Table 3 medicina-59-01452-t003:** Subgroup analysis of estimated GATA3 positive rates based on various subtypes of urothelial carcinomas.

	Number of Subsets	Fixed Effect [95% CI]	Heterogeneity Test (*p*-Value)	Random Effect [95% CI]	Egger’s Test (*p*-Value)
Noninvasive UC *	11	0.969 [0.958, 0.977]	0.007	0.965 [0.938, 0.980]	0.586
Carcinoma in situ	2	0.961 [0.916, 0.982]	0.250	0.956 [0.878, 0.985]	-
Invasive UC	55	0.626 [0.608, 0.643]	<0.001	0.644 [0.581, 0.702]	0.474
Non-muscular invasion ^#^	6	0.941 [0.902, 0.965]	0.259	0.937 [0.883, 0.967]	0.419
Muscular invasion	13	0.720 [0.685, 0.752]	<0.001	0.753 [0.645, 0.836]	0.400
Histologic subtypes					
Adenocarcinoma	3	0.190 [0.093, 0.350]	0.647	0.190 [0.093, 0.350]	0.154
Clear cell	1	0.929 [0.423, 0.996]	1.000	0.929 [0.423, 0.996]	-
Glandular differentiation	2	0.474 [0.268, 0.689]	0.809	0.474 [0.268, 0.689]	-
Lymphoepithelioma-like	1	0.300 [0.100, 0.624]	1.000	0.300 [0.100, 0.624]	-
Microcystic	2	0.952 [0.724, 0.993]	0.463	0.952 [0.724, 0.993]	-
Micropapillary	5	0.773 [0.697, 0.834]	<0.001	0.862 [0.661, 0.952]	0.264
Nested	1	0.700 [0.376, 0.900]	1.000	0.700 [0.376, 0.900]	-
Plasmacytoid	4	0.756 [0.637, 0.845]	0.003	0.825 [0.517, 0.954]	0.544
Pleomorphic giant cell	1	0.909 [0.561, 0.987]	1.000	0.909 [0.561, 0.987]	-
Sarcomatoid	8	0.385 [0.316, 0.459]	0.003	0.407 [0.282, 0.545]	0.370
Small cell neuroendocrine carcinoma	4	0.132 [0.059, 0.267]	0.310	0.125 [0.051, 0.276]	0.231
Squamous cell carcinoma	2	0.172 [0.069, 0.367]	0.220	0.141 [0.031, 0.454]	-
Squamous differentiation	3	0.281 [0.159, 0.447]	0.284	0.258 [0.122, 0.467]	0.003
Signet ring cell carcinoma	1	0.409 [0.228, 0.618]	1.000	0.409 [0.228, 0.618]	-
Undifferentiated	1	0.643 [0.376, 0.843]	1.000	0.643 [0.376, 0.843]	-

CI, confidence interval; UC, urothelial carcinoma. *, *p* < 0.001 in the meta-regression test between noninvasive and invasive urothelial carcinoma subgroups. ^#^, *p* = 0.041 in the meta-regression test between non-muscular and muscular invasion subgroups.

## Data Availability

Not applicable.

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
