# Peer review of "Diagnostic and Prognostic Roles of GATA3 Immunohistochemistry in Urothelial Carcinoma"

_medicina, 2023, doi:10.3390/medicina59081452_

Round 1

Reviewer 1 Report

Yoo et al, in their article titled “Diagnostic and prognostic roles of GATA3 immunohistochemistry in urothelial carcinoma” sought to evaluate the role of GATA3 as a diagnostic and prognostic factor in urothelial carcinoma (UC) using meta-analysis. Authors based on GATA3 expression rates in various tumor tissue samples conclude that GATA3 expression significantly correlates with low-stage and better prognosis in UC. Authors have nicely written the article but need improvements at various places.

Comments:

1.       At line 19, “GATA expression” is it GATA3 expression you mean?

2.       GATA3 should be given in full form in Abstract at line 13.

3.       Abbreviation should be given in full in the first place of occurrence only.

4.       The sent starting at line 83 which reads as “As the eligible studies used various antibodies….” makes poor sense, authors should re-write this sent for more clarity.

5.       Statistical analyses should be provided in more detail.

6.       What does “The expression rates of UC…” at line 105 mean? Do you mean GATA3 instead of UC?

7.       Text at line 132 is not clear.

8.       As per Table 3 and starting at line 21 “In invasive UCs, there was a significant difference in GATA3 expression between non-muscular invasion and muscular invasion subgroups (0.937, 95% CI: 0.883–0.967 vs. 0.753, 95% CI: 0.645– 22 0.836)” and result given at line 114 doesn’t match OR the results are not properly described. Authors need to make text clearer to the general readership.

9.       GATA3 expression and survival rate correlation should be presented with more clarity. Present figure 2 is not much informative and visual.  Graph should show y-axis and x-axis with indicated values and labels.  

10.   Authors should make discussion more clear at places where they discuss UC.

11.   GATA3 role should be discussed in general (1/2 para in start of discussion).

12.   Authors could discuss GATA3 and its impact on tumor immunity and correlate their findings with anti-tumor effects and possible immunotherapy.

13.   References should properly checked for errors.

English can be improved. Some sentences are poorly written. 

Author Response

Yoo et al, in their article titled “Diagnostic and prognostic roles of GATA3 immunohistochemistry in urothelial carcinoma” sought to evaluate the role of GATA3 as a diagnostic and prognostic factor in urothelial carcinoma (UC) using meta-analysis. Authors based on GATA3 expression rates in various tumor tissue samples conclude that GATA3 expression significantly correlates with low-stage and better prognosis in UC. Authors have nicely written the article but need improvements at various places.

Comments:

  1. At line 19, “GATA expression” is it GATA3 expression you mean?

Response) As a recommendation, we corrected.

  1. GATA3 should be given in full form in Abstract at line 13.

Response) As a recommendation, we added the full name.

  1. Abbreviation should be given in full in the first place of occurrence only.

Response) As a recommendation, we checked and corrected.

  1. The sent starting at line 83 which reads as “As the eligible studies used various antibodies….” makes poor sense, authors should re-write this sent for more clarity.

Response) As a recommendation, we corrected the sentence.

  1. Statistical analyses should be provided in more detail.

Response) As a recommendation, we added the description in the revised manuscript.

  1.  What does “The expression rates of UC…” at line 105 mean? Do you mean GATA3 instead of UC?

Response) As a recommendation, we corrected from UC to GATA3.

  1.  Text at line 132 is not clear.

Response) As a recommendation, we corrected the sentence in the revised manuscript.

  1. As per Table 3 and starting at line 21 “In invasive UCs, there was a significant difference in GATA3 expression between non-muscular invasion and muscular invasion subgroups (0.937, 95% CI: 0.883–0.967 vs. 0.753, 95% CI: 0.645– 22 0.836)” and result given at line 114 doesn’t match OR the results are not properly described. Authors need to make text clearer to the general readership.

Response) We corrected the sentence in the result (Line 114) as follow: GATA3 expression was significantly lower in non-muscular-invasive UC than in non-muscular-invasive UC (0.753, 95% CI: 0.645–0.836 vs. 0.937, 95% CI: 0.883–0.967; p = 0.041 in the meta-regression test). In addition, this result is not OR result. This result is a representation of the GATA3 expression rate.

  1. GATA3 expression and survival rate correlation should be presented with more clarity. Present figure 2 is not much informative and visual.  Graph should show y-axis and x-axis with indicated values and labels.  

Response) Unfortunately, Figure 2 is an uneditable forest plot provided by the CMA program. In addition, a new estimated value is obtained through a meta-analysis. That is, the correlation between GATA3 and survival is interpreted by the obtained HR. To clarify the correlation, we added the explanation in the revised manuscript.

  1. Authors should make discussion more clear at places where they discuss UC.

Response) As a recommendation, we added the discussion in the revised manuscript.

  1. GATA3 role should be discussed in general (1/2 para in start of discussion).

Response) As a recommendation, we added the discussion in the revised manuscript.

  1. Authors could discuss GATA3 and its impact on tumor immunity and correlate their findings with anti-tumor effects and possible immunotherapy.

Response) As a recommendation, we added the discussion in the revised manuscript as follow: In non-muscle invasive UC, intravesical immunotherapy with Bacillus Calmette-Guérin (BCG) is used to reduce recurrence [60]. However, despite the use of BCG, tumor recurrence is common. Group 2 innate lymphoid cells (ILC2) express high levels of the transcription factor GATA3 [60]. As ILC2 may be the immunotherapeutic target in UC, elucidation of the role of GATA3 may be important.

Reference.

Tumino, N.; Vacca, P.; Quatrini, L.; Munari, E.; Moretta, F.; Pelosi, A.; Mariotti, F.R.; Moretta, L. Helper Innate Lymphoid Cells in Human Tumors: A Double-Edged Sword? Front Immunol 2020, 10, 3140.

  1. References should properly checked for errors.

Response) As a recommendation, we checked the references.

Reviewer 2 Report

Authors aimed to investigate diagnostic and prognostic roles of GATA3 immunohistochemistry in urothelial carcinoma. For this purpose, they performed a meta-analysis and investigated GATA3 immunohistochemical expression rates and performed a subgroup analysis based on tumor site, study location, and histological subtypes. Authors found a significant correlation between GATA3 expression and better prognosis (hazard ratio: 0.402, 95% CI: 0.311–0.521), as GATA3 expression significantly correlated with low-stage and better prognosis in UC. However, GATA3 expression is highly variable across histological subtypes and it is careful interpreting GATA3 expression. In summary, authors need to be congratulated on the study, it covers a very relevant topic. Introduction is reasonable, stats and concomitant results are sound. The discussion is coherent and interprets the findings with current literature. However, some details need to be addressed to strengthen the overall manuscript. In light of limited reliable biomarkers for survival of BCa patients, this study proposes a novel, and interesting, new possibility. However, there was a recent study following-up a large patient cohort (>1,000), assessing hemoglobin, and C-reactive protein for survival in patients with BCa who underwent RC (PMID: 32114587). In light of your interesting findings, how would you comment on that study? Additionally, include in introduction: l. 37-38, PMID: 32114587. However, there is evidence that RC is still feasible even in the oldest-old patient cohort (>/= 85 years of age). This was also recently demonstrated (PMID: 32871580), and at least the introduction would profit from including this. Additionally, and regarding PMID: 32871580, have you observed any differences, or correlation in GATA3 expression when stratifying patients by age?

Quality of English language is fine.

Author Response

Authors aimed to investigate diagnostic and prognostic roles of GATA3 immunohistochemistry in urothelial carcinoma. For this purpose, they performed a meta-analysis and investigated GATA3 immunohistochemical expression rates and performed a subgroup analysis based on tumor site, study location, and histological subtypes. Authors found a significant correlation between GATA3 expression and better prognosis (hazard ratio: 0.402, 95% CI: 0.311–0.521), as GATA3 expression significantly correlated with low-stage and better prognosis in UC. However, GATA3 expression is highly variable across histological subtypes and it is careful interpreting GATA3 expression. In summary, authors need to be congratulated on the study, it covers a very relevant topic. Introduction is reasonable, stats and concomitant results are sound. The discussion is coherent and interprets the findings with current literature. However, some details need to be addressed to strengthen the overall manuscript.

In light of limited reliable biomarkers for survival of BCa patients, this study proposes a novel, and interesting, new possibility. However, there was a recent study following-up a large patient cohort (>1,000), assessing hemoglobin, and C-reactive protein for survival in patients with BCa who underwent RC (PMID: 32114587). In light of your interesting findings, how would you comment on that study? Additionally, include in introduction: l. 37-38, PMID: 32114587.

Response) As a recommendation, the paper is about a laboratory parameter and the results are quite interesting. These results may be complementary with our results, which are obtained from human tissues. We added the brief results of above reference in the revised manuscript.

However, there is evidence that RC is still feasible even in the oldest-old patient cohort (>/= 85 years of age). This was also recently demonstrated (PMID: 32871580), and at least the introduction would profit from including this. Additionally, and regarding PMID: 32871580, have you observed any differences, or correlation in GATA3 expression when stratifying patients by age?

Response) As a recommendation, we added the comment for the importance of patient’s age in UC using above reference. However, the detailed analysis based on patients’ age could not be performed due to insufficient information of eligible studies. We described the limitation of additional analysis in the revised manuscript.